# Identification and Genome Characterization of Begomovirus and Satellite Molecules Associated with Lettuce (*Lactuca sativa* L.) Leaf Curl Disease

**DOI:** 10.3390/plants14050782

**Published:** 2025-03-04

**Authors:** Yafei Tang, Mengdan Du, Zhenggang Li, Lin Yu, Guobing Lan, Shanwen Ding, Tahir Farooq, Zifu He, Xiaoman She

**Affiliations:** 1Guangdong Provincial Key Laboratory of High Technology for Plant Protection, Plant Protection Research Institute, Guangdong Academy of Agricultural Sciences, Guangzhou 510640, China; tangyafei@gdppri.com (Y.T.); go_ahead0220@163.com (M.D.); lizhenggang@gdppri.com (Z.L.); yulin@gdppri.com (L.Y.); languobing@gdppri.com (G.L.); dingshanwen@gdppri.com (S.D.); tahirfarooq@gdppri.com (T.F.); hezf@gdppri.com (Z.H.); 2South China Key Laboratory for Green Prevention and Control of Fruits and Vegetables, Ministry of Agriculture and Rural Affairs, Guangzhou 510640, China; 3College of Plant Protection, South China Agricultural University, Guangzhou 510640, China

**Keywords:** lettuce leaf curl disease, pepper leaf curl Yunnan virus, tomato leaf curl China betasatellite, ageratum yellow vein alphasatellite, new geminialphasatellite

## Abstract

Lettuce (*Lactuca sativa* L.) plants showing leaf curl and vein enation symptoms were found in Yunnan province, China. PCR detection with genus-specific primers revealed that symptomatic lettuce plants were infected with *Begomovirus*. The full-length viral component and satellite molecules were obtained by RCA, restriction enzyme digestion, PCR, cloning and DNA sequencing. A viral component (YN-2023-WJ) and three satellite molecules (YN-2023-WJ-alpha1, YN-2023-WJ-alpha2 and YN-2023-WJ-beta) were obtained from diseased lettuce plants. YN-2023-WJ exhibited the highest nt identity at 97.1% with pepper leaf curl Yunnan virus isolated from cigar plants. YN-2023-WJ-beta displayed the highest nt identity at 93.9% with tomato leaf curl China betasatellite. YN-2023-WJ-alpha1 showed the highest nt identity at 94.7% with ageratum yellow vein alphasatellite. YN-2023-WJ-alpha2 shared the highest nt identity at 75.6% with gossypium mustelinum symptomless alphasatellite and vernonia yellow vein Fujian alphasatellite. Based on the threshold for the classification of *Begomovirus*, *Betasatellite* and *Alphasatellite*, YN-2023-WJ was designated as a new isolate of PepLCYnV, YN-2023-WJ-beta as a new isolate of ToLCCNB and YN-2023-WJ-alpha1 as a new member of AYVA, whereas YN-2023-WJ-alpha2 was identified as a new geminialphasatellite species, for which the name pepper leaf curl Yunnan alphasatellite (PepLCYnA) is proposed. To the best of our knowledge, this is the first report of *L. sativa* L. infection by PepLCYnV associated with ToLCCNB, AYVA and PepLCYnA, and *L. sativa* L. is a new host plant of *Begomovirus*.

## 1. Introduction

Plant viruses, especially members of the *Geminiviridae* family, are important worldwide pathogens that cause devastating yield losses in many economically important crops, such as tomato, cotton, cassava, corn and beans. These viruses have a small circular, single-stranded DNA genome encapsidated in twinned, icosahedral particles. According to the latest report of the International Committee on Taxonomy of Viruses (ICTV) in 2023 (https://ictv.global/taxonomy) (accessed on 10 October 2024), the *Geminiviridae* family includes 522 species of viruses from 15 genera, namely *Becurtovirus* (3 species), *Begomovirus* (445 species), *Capulavirus* (4 species), *Citlodavirus* (4 species), *Curtovirus* (3 species), *Eragrovirus* (1 species), *Grablovirus* (3 species), *Maldovirus* (3 species), *Mastrevirus* (45 species), *Mulcrilevirus* (2 species), *Opunvirus* (1 species), *Topilevirus* (2 species), *Topocuvirus* (1 species), *Turncurtovirus* (1 species) and *Welwivirus* (2 species). Thus, the genus *Begomovirus* is the largest group within this family, and most importantly, it causes yield losses in many crops, and is widely distributed in tropical and subtropical regions of the world. The viruses included in the *Begomovirus* genus are transmitted exclusively by the whitefly (*Bemisia tabaci*), generally referred to as whitefly-transmitted geminiviruses [1,2].

Begomoviruses are either monopartite or bipartite viruses depending on the number of genome components. The monopartite begomoviruses have only the DNA-A component, while the bipartite consist of two components, referred to as DNA-A and DNA-B, each of 2.5–2.6 kb [3,4]. The organization of genes or open reading frames (ORFs) in the genome of monopartite begomoviruses resembles that of the bipartite DNA-A component [5,6], whereas the DNA-B encodes for only two ORFs, the first of which is a nuclear shuttle protein (BV1) on the virion-sense strand and a movement protein (BC1) on the complementary-sense strand [7,8]. Some begomoviruses are associated with circular, single-stranded DNA (ssDNA) satellites, namely betasatellites (genus *Betasatallite*, family, *Tolecusatellitidae*), alphasatellites (family *Alphasatellitidae*) and deltasatellites (genus *Deltasatellite*, family *Tolecusatellitidae*) [9,10,11,12]. They are approximately 1.3 kb (betasatellites and alphasatellites) or 0.7 kb (deltasatellites). All satellites share no significant sequence similarity with the viral components.

Lettuce (*Lactuca sativa* L.), an *Asteraceae* family member in the *Lactuca* genus, is an annual or biennial herbaceous plant. It is a popular vegetable and an economically important crop native to the Mediterranean region, primarily cultivated across Europe, the Americas and Asia [13]. Viral diseases are common in lettuce production and often cause severe losses [14]. Several viruses have been reported to infect lettuce, including alfalfa mosaic virus (AMV) [15], artichoke yellow ringspot virus (AYRSV) [16], broad bean wilt virus (BBWV) [15], cucumber mosaic virus (CMV) [15], lettuce big-vein-associated virus (LBVAV) [17], lettuce chlorosis virus (LCV) [18], lettuce mosaic virus (LMV) [19], lettuce necrotic yellows virus (LNYV) [20], lettuce Italian necrotic virus (LINV) [21], lettuce virus X (LeVX) [22], mirafiori lettuce big-vein virus (MLBVV) [17,23], tomato spotted wilt virus (TSWV) [24], turnip mosaic virus (TuMV) [25,26], turnip yellow virus (TuYV) [15], tobacco streak virus (TSV) [27], etc. However, to date, no begomovirus has been reported to infect lettuce.

In April 2023, lettuce plants exhibiting leaf curl and vein enation symptoms were found in Yuxi city of Yunnan province, China. To identify the causative begomovirus, we employed rolling circle amplification (RCA), restriction enzyme digestion, polymerase chain reaction (PCR), cloning and subsequent DNA sequencing techniques, followed by sequence analyses for classification. Our findings represent the first discovery of lettuce as a new natural host of *Begomovirus* in China, along with the molecular characteristics of a begomovirus and three satellite molecules, including a new geminialphasatellite, isolated from lettuce.

## 2. Results

### 2.1. Field Survey and Begomovirus Detection

In April 2023, some abnormal lettuce plants, suspected to be infected by *Begomovirus*, were found on a farm situated in Yuxi city of Yunnan province, China. These diseased plants exhibited leaf curl and vein enation symptoms (Figure 1). The incidence was about 80%.

Ten symptomatic and one asymptomatic lettuce sample were detected by PCR using the *Begomovirus* degenerate primer pair AV_494_/CoPR. An amplicon of the expected size (~570 bp) was obtained from all symptomatic samples, whereas no fragment was obtained from the asymptomatic sample (Figure 2). The PCR result indicated that all symptomatic samples were infected by *Begomovirus*.

### 2.2. Sequence Analysis of the Begomoviral Genome and Satellite DNA

The begomoviral genome from the diseased lettuce plants was obtained from the RCA products digested by BamH I. The viral genome had typical features of monopartite begomoviruses, was 2749 nt in size and encoded six putative ORFs, including the *V*1 (291–1061 nt) and *V*2 (131–478 nt) genes in the virion sense strand, *C*1 (1510–2598 nt), *C*2 (1203–1610 nt), *C*3 (1058–1462 nt) and *C*4 (2151–2441 nt) genes in the complementary sense strand, an intergenic region (IR) containing a conserved nonanucleotide sequence (TAATATTAC) and a potential stem-loop structure around this nonanucleotide motif.

BLASTn results showed that the complete begomoviral genome sequence shared a high nt identity with many begomoviruses. Furthermore, according to the SDT analysis, the pairwise nt sequence identities ranged from 81.2 to 97.1% for the viral component sequence and for 31 other closely related begomoviruses available in the GenBank release 262 (Figure 3). The viral component sequence was most closely related to the pepper leaf curl Yunnan virus (PepLCYnV) isolates, which shared >92.9% nt identity with all available PepLCYnV isolates, with the highest nt identity at 97.1% belonging to the YN-2023 isolate, which was isolated from cigar plants (GenBank accession no. OR756247). Then, it shared 84.1~87.4% nt identity with the related isolates of tomato yellow leaf curl China virus (TYLCCNV), and 81.2~86.2% nt identity with the related isolates of ageratum yellow vein China virus (AYVCNV). Phylogenetic analysis revealed that the isolate in this study clustered with all available PepLCYnV isolates to form a unique clade (Figure 4). Based on the new demarcation thresholds for *Begomovirus*, the new isolate should be classified as belonging to the strain with which the full-length genome or DNA-A component shares ≥94% nt sequence identity with any one isolate from that strain of that species [4]. Therefore, the isolate in this study was designated as a new isolate of PepLCYnV, named as YN-2023-WJ isolate, and submitted to the National Center for Biotechnology Information (NCBI) to obtain the GenBank accession number PQ352196.

Two satellite molecules from diseased lettuce plants were obtained from the RCA products digested by BamH I and EcoR I. Two satellite molecules (named YN-2023-WJ-alpha1 and YN-2023-WJ-alpha2) had typical features of geminialphasatellites, consisting of a single conserved Rep encoding gene in the virion-sense, and a predicted hairpin structure at the presumed origin of virion strand replication that contained a TAGTATAC loop sequence. YN-2023-WJ-alpha1 contained 1364 nt, and a single conserved rep encoding gene (74–1021 nt). YN-2023-WJ-alpha2 contained 1366 nt, a single conserved rep encoding gene (77–1003 nt). According to the SDT analysis, the pairwise nt sequence identity between two satellite molecules was 67.9%. YN-2023-WJ-alpha1 shared the highest nt identity at 94.7% with ageratum yellow vein alphasatellite (AYVA-pAYVV1/7, GenBank accession no. NC_039080). YN-2023-WJ-alpha2 shared the highest nt identities at 75.6% with the gossypium mustelinum symptomless alphasatellite (GMusSLA-Gossyp-4, GenBank accession no. FJ218494) and 75.6% with the vernonia yellow vein Fujian alphasatellite (VeYVFA, GenBank accession no. JF733780). Using the banana bunchy top alphasatellite (GenBank accession no. L32167, belonging to the *Nanoalphasatellitinae* subfamily) as an outgroup, phylogenetic analysis between two satellite molecules and other geminialphasatellites was performed. The tree showed that YN-2023-WJ-alpha1 closely clustered with AYVA isolates, and YN-2023-WJ-alpha2 clustered with other isolates of vernonia yellow vein Fujian alphasatellite, mesta yellow vein mosaic alphasatellite, okra enation leaf curl alphasatellite, gossypium mustelinum symptomless alphasatellite and gossypium davidsonii symptomless alphasatellite, whereas it formed a separate branch and exhibited a relatively distant evolutionary distance from these isolates (Figure 5). Based on the species (88% nt identity) demarcation threshold for geminialphasatellites [9], YN-2023-WJ-alpha1 was a new member of AYVA, and YN-2023-WJ-alpha2 was a new geminialphasatellite species, for which the name pepper leaf curl Yunnan alphasatellite (PepLCYnA) was proposed. Two complete sequences of satellite molecules were submitted to the National Center for Biotechnology Information (NCBI) to obtain the GenBank accession numbers PQ352197 and PQ352199.

Another satellite molecule from diseased lettuce plants was obtained via PCR amplification with β01/β02 primers. The satellite molecule (YN-2023-WJ-beta) with a 1347 nt genome shared typical features of DNA β associated with *Begomovirus*. Further, it consisted of a region with an adenine-rich (A-rich) sequence, the satellite conserved region (SCR), a single ORF (βC1, 210–566 nt) in the complementary-sense strand and a single ORF (βV1, 283–678 nt) in the virion-sense strand [28]. According to SDT analysis, YN-2023-WJ-beta was most closely related to tomato leaf curl China betasatellite (ToLCCNB) isolates, and shared >78.0% nt identity with the available ToLCCNB isolates, with the highest nt identity at 93.9% with ToLCCNB-Y4536 (GenBank accession no. KT995473). Based on the species demarcation threshold for distinct DNA β [29], YN-2023-WJ-beta was a new isolate of ToLCCNB, and was submitted to the National Center for Biotechnology Information (NCBI) to obtain the GenBank accession number PQ352198.

### 2.3. Specific Detection of Virus and Satellite Molecules in Symptomatic Samples

Using the specific primers PepLCYnV-F/PepLCYnV-R, alpha1-F/alpha1-R, alpha2-F/alpha2-R and beta-F/beta-R to detect the presence of PepLCYnV, AYVA, PepLCYnA and ToLCCNB, respectively, the expected fragments were amplified from all ten symptomatic lettuce samples. These results confirmed that these diseased lettuce samples were co-infected by PepLCYnV, AYVA, PepLCYnA and ToLCCNB (Figure 6).

## 3. Discussion

Lettuce is an important vegetable containing abundant vitamin C, phenolic compounds and fiber content [15]. Virus infections are known to induce symptoms such as mosaic, mottle, necrotic yellows, vein clearing and shrinking in lettuce. Lettuce leaf curl disease, exhibiting leaf curl and vein enation symptoms, was first discovered in China.

A full-length viral component and three satellite molecules linked to lettuce leaf curl disease in China were obtained by the enrichment of circular DNA with rolling circle amplification, restriction enzyme digestion, PCR, cloning and DNA sequencing. Based on the threshold for the classification of *Begomovirus*, *Betasatellite* and *Alphasatellite*, the virus and three satellite molecules were PepLCYnV, ToLCCNB, AYVA and PepLCYnA, respectively. PepLCYnA, an undiscovered alphasatellite, was also found. This result will provide materials for the alphasatellite species diversity.

PepLCYnV (GenBank accession no. NC_010618) was first identified as a novel begomovirus infecting *Capsicum annuum* in Yunnan, China, and it was associated with betasatellite (PepLCYnB, GenBank accession no. NC_010619). According to the GenBank database re, thus far, this virus has occurred in China, and in naturally infected *Capsicum annuum*, cigar, tomato and *Clerodendrum bungei.* Here, PepLCYnV and three other satellite molecules (AYVA, PepLCYnA, ToLCCNB) were detected in diseased lettuce samples, but PepLCYnB was not detected. Therefore, PepLCYnV can co-infect host plants associated with heterologous satellite molecules. This is the first report of *L. sativa* L. infection by PepLCYnV associated with ToLCCNB, AYVA and PepLCYnA, and *L. sativa* L. is a new host plant of *Begomovirus.*

*Begomovirus* is the largest genus, including 445 virus species that account for approximately 85% of all geminiviruses reported in the world. Mixed infections of begomoviruses have been frequently documented in various crops, including *Solanaceous* crops and cucurbits across different regions [30,31,32]. In China, 86 begomovirus species were reported [33]. Mixed infections of begomoviruses are common. For instance, more than 20 begomovirus species have been reported to infect tomato plants in field conditions [34]. In the current study, only PepLCYnV was detected in *L. sativa* L. samples. To future explore the potential presence of other begomoviruses, future research will employ additional methodologies, such as digestion with more restriction enzymes, PCR amplification using specific primers targeting other viruses or deep sequencing techniques to comprehensively detect and identify additional viral species in *L. sativa* L. plants.

## 4. Materials and Methods

### 4.1. Field Investigation and Sample Collection

During a routine survey in April 2023, lettuce plants exhibiting leaf curl and vein enation symptoms were found in Yuxi city of Yunnan province, China (24°4′1″ N, 101°33′59″ E). Based on symptomatology, we suspected that this disease may have been caused by *Begomovirus*. Ten samples were randomly collected from different diseased plants, while one sample was collected from asymptomatic plants.

### 4.2. Begomovirus Detection

Total DNA was extracted from each sample separately using the EasyPure Plant DNA kit (TransGen Biotech, Beijing, China). PCR detection was performed with *Begomovirus* universal primers AV_494_(5′-GCCYATRTAYAGRAAGCCMAG-3′)/CoPR (5′-GANGSATGHGTRCADGCCATATA-3′) [35,36] targeting the partial *AV1* gene. The amplification protocols were 94 °C for 10 min; 35 cycles each of 94 °C for 30 s, 52 °C for 30 s and 72 °C for 30 s; and a final extension at 72 °C for 10 min. PCR products were analyzed by electrophoresis on a 1% agarose gel, visualized under a UV transilluminator.

### 4.3. Cloning of the Begomoviral Genome and Satellite Molecules

A positive sample was randomly selected for amplification using a rolling circle amplification kit (GE Healthcare, Buckinghamshire, United Kingdom), and digested with BamH I and EcoR I, respectively. The ~2.7 kbp target fragments of the putative begomoviral genome and ~1.5 kbp target fragments of a putative satellite molecule were separated by 1% agarose gel electrophoresis in 1× Tris acetic acid–ethylenediaminetetraacetic acid buffer (pH 8.0), gel-purified, individually cloned into pGEM-3Z vector and sequenced (Invitrogen Co., Shanghai, China). Betasatellite was obtained by PCR amplification with β01/β02 primers for betasatellites [37]. The target fragment was then cloned into pMD19T vector and sequenced (Sangon Biotech, Shanghai, China).

### 4.4. Sequence Analysis

The DNA sequences were assembled, edited, and analyzed using DNAStar software version 5.0 (DNAStar Inc., Madison, WI, USA). The preliminary identity of the obtained DNA sequences was established via BLASTn analysis against the sequences available in the GenBank database (www.ncbi.nlm.nih.gov) (accessed on 17 Auguest 2024). The pairwise nucleotide (nt) sequence identity was determined by MUSCLE alignment using the Sequence Demarcation Tool (SDT 1.2) [38]. The phylogenetic tree was constructed by the neighbor-joining method with 1000 bootstrap values using MEGA 6.0 [39].

### 4.5. PCR Detection for Virus and Satellite Molecules

According to the obtained sequences, the specific primer pairs PepLCYnV-F/PepLCYnV-R, alpha1-F/alpha1-R, alpha2-F/alpha2-R and beta-F/beta-R were designed to amplify the 729 bp fragment of PepLCYnV, 527-bp fragment of AYVA, 797 bp fragment of PepLCYnA and 641 bp fragment of ToLCCNB (Table 1), and to detect the presence of PepLCYnV, AYVA, PepLCYnA and ToLCCNB in the diseased samples, respectively. The amplification protocols were 94 °C for 10 min; 35 cycles each of 94 °C for 30 s, 54 °C or 53 °C for 30 s and 72 °C for 1 min; and a final extension at 72 °C for 10 min. PCR products were analyzed by electrophoresis on 1% agarose gel and visualized under a UV transilluminator. Then, each target fragment was sequenced (Sangon Biotech, Shanghai, China).

## 5. Conclusions

In the current study, lettuce leaf curl disease was first discovered in China. We found PepLCYnV and three satellite molecules (ToLCCNB, AYVA and PepLCYnA) complexes to be linked to lettuce leaf curl disease in China. The characterization and phylogenetic relationship between PepLCYnV, ToLCCNB, AYVA and PepLCYnA associated with lettuce leaf curl disease in China were analyzed. To the best of our knowledge, this is the first report of *L. sativa* L. infection by PepLCYnV associated with ToLCCNB, AYVA and PepLCYnA, and *L. sativa* L. is a new host plant of *Begomovirus*. In addition to PepLCYnA, undiscovered alphasatellites were also found. This study focuses on the identification the types of viruses and satellite molecules infecting lettuce in China. The results will provide a theoretical basis for the prevention and control of lettuce leaf curl disease, and provide materials for the alphasatellite species diversity.

## Figures and Tables

**Figure 1 plants-14-00782-f001:**
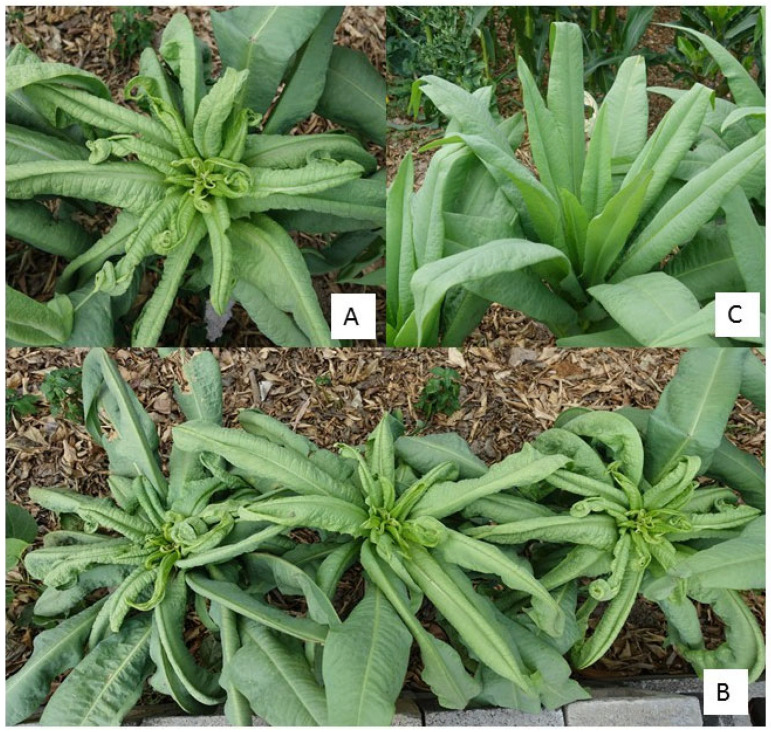
Symptoms of diseased *Lactuca sativa* L. plants exhibiting leaf curling and vein enation symptoms. (**A**,**B**) Diseased plants and (**C**) healthy plant.

**Figure 2 plants-14-00782-f002:**
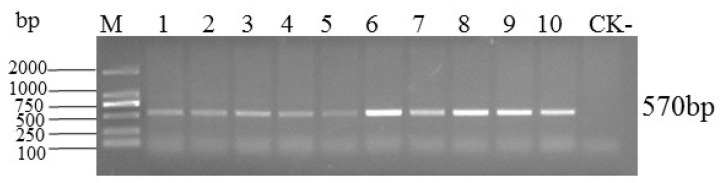
Infection of *Begomovuris* PCR detection in *Lactuca sativa* L. M: DL2000 DNA marker, 1–10: diseased *Lactuca sativa* L. samples, CK-: the asymptomatic sample.

**Figure 3 plants-14-00782-f003:**
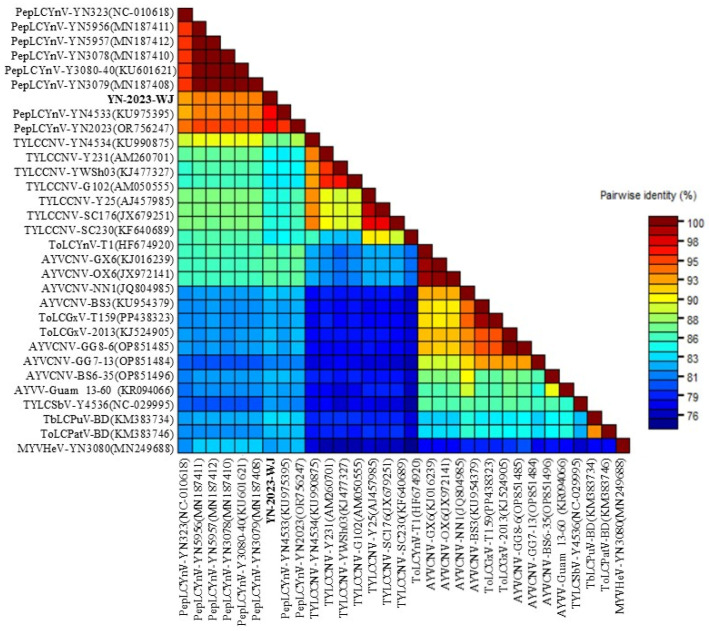
The pairwise nucleotide (nt) sequence identity between YN-2023-WJ and the other 31 isolates with MUSCLE alignment using the Sequence Demarcation Tool (SDT 1.2). PepLCYnV: pepper leaf curl Yunnan virus, TYLCCNV: tomato yellow leaf curl China virus, AYVCNV: ageratum yellow vein China virus, ToLCYnV: tomato leaf curl Yunnan virus, ToLCGxV: tomato leaf curl Guangxi virus, AYVV: ageratum yellow vein virus, TbLCPuV: tobacco leaf curl Pusa virus, ToLCPatV: tomato leaf curl Patna virus, TYLCSbV: tomato yellow leaf curl Shuangbai virus and MYVHeV: Malvastrum yellow vein Honghe virus.

**Figure 4 plants-14-00782-f004:**
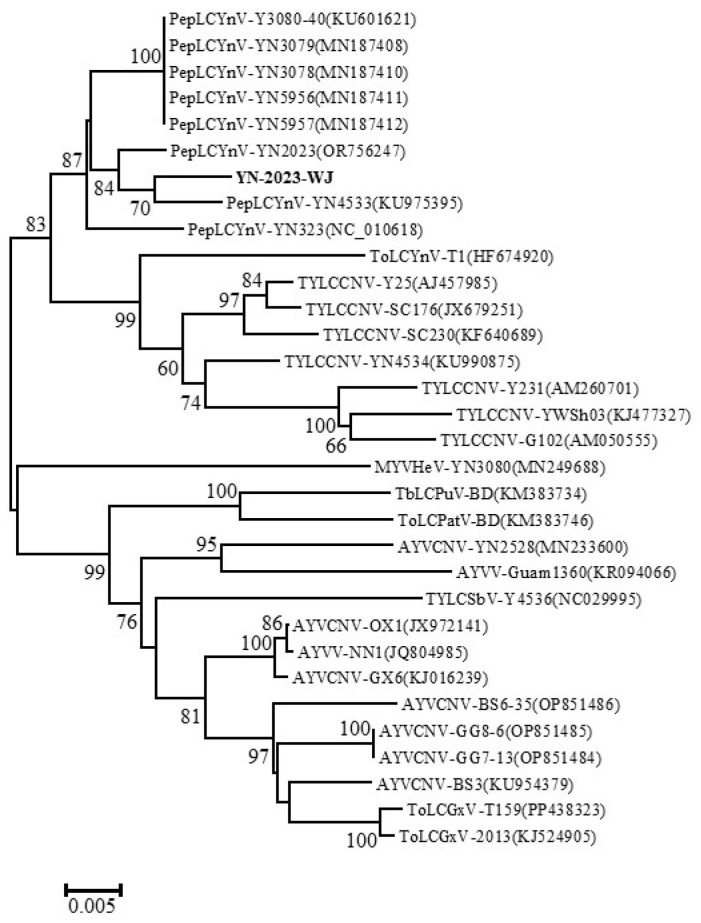
Phylogenetic tree of YN-2023-WJ and 31 other isolates based on the complete genome sequence. Bootstrap analysis was repeated 1000 times to evaluate the significance of the internal branches. Bootstrap values below 50% are not shown.

**Figure 5 plants-14-00782-f005:**
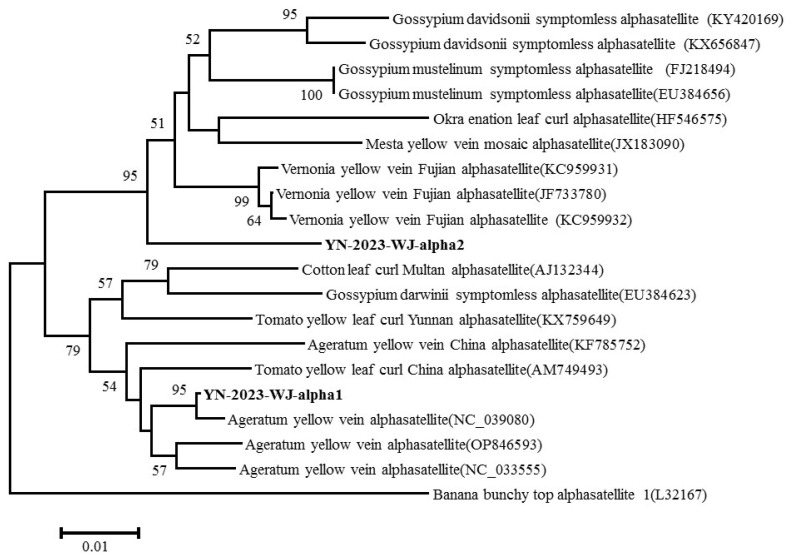
Phylogenetic tree of the two alphasatellites identified in this study (YN-2023-WJ-alpha1 and YN-2023-WJ-alpha2) and other geminialphasatellites based on the complete genome sequence using banana bunchy top alphasatellite (GenBank accession no. L32167, belonging to the Nanoalphasatellitinae subfamily) as an outgroup. Bootstrap analysis was repeated 1000 times to evaluate the significance of the internal branches. Bootstrap values below 50% are not shown.

**Figure 6 plants-14-00782-f006:**
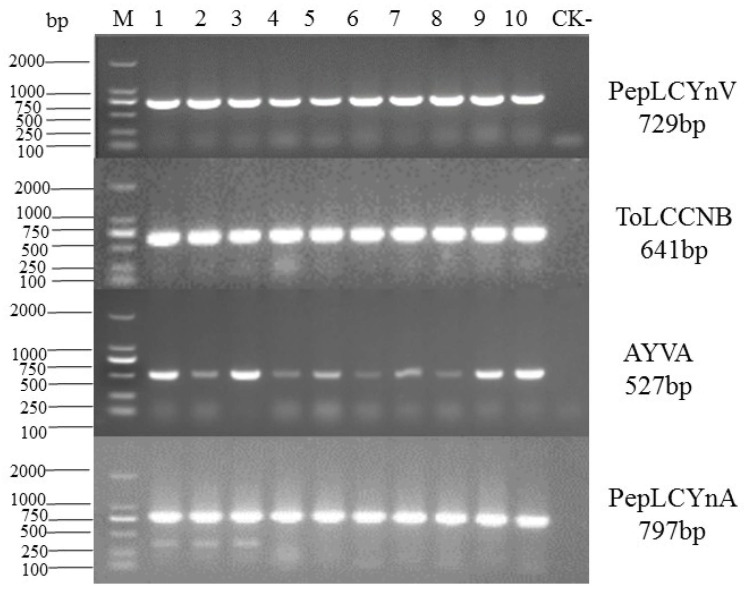
Verification of PepLCYnV, TYLCCNB, AYVA and PepLCYnA infection in *Lactuca sativa* L. using PCR. M: DL2000 DNA marker, 1–10: diseased *Lactuca sativa* L. samples, CK-: the asymptomatic sample.

**Table 1 plants-14-00782-t001:** List of primers used in this study.

Primer Name	Primer Sequence (5′–3′)	Target	Annealing Temperature (°C)	Size (bp)
PepLCYnV-F	TGCCAGGGATTATGTCGAAG	PepLCYnV	54	729
PepLCYnV-R	CAGGATTACTCGCATGAGTAG			
beta-F	CCCCTACATCTATATCTTCTACTG	ToLCCNB	53	641
beta-R	GCGCTCCCTTTTGTTTCTTAAA			
alpha1-F	TTTCACCGTCTTCTTCCTTTCTG	AYVA	53	527
alpha1-R	GACCATACACCCAGAAGATAGTG			
alpha2-F	GGCTGCCCTTAAGAGTGTAT	PepLCYnA	53	797
alpha2-R	CTAGGGCATCAATAAACCCAAC			

## Data Availability

I have mentioned the accession number in manuscript which is already available in the NCBI database.

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
