# Peer review of "Identification and Genome Characterization of Begomovirus and Satellite Molecules Associated with Lettuce (Lactuca sativa L.) Leaf Curl Disease"

_plants, 2025, doi:10.3390/plants14050782_

Round 1
Reviewer 1 Report
Comments and Suggestions for Authors
In this manuscript, the authors proposed the full-length viral component and three satellite molecules (PepLCYnV, ToLCCNB) linked to lettuce-infecting Begomovirus first reported in Yunnan, China. Methodology essentially included enrichment of circular DNA by RCA, followed by restriction enzyme digestion, PCR, cloning, and DNA sequencing. Meanwhile, PepLCYnA, an undiscovered alphasatellite was also proposed. It was believed to be the first discovery of lettuce as a new natural host of Begomovirus in China.
Understanding the genetic makeup of plant viruses and satellite DNAs could provide imminent opportunities for harnessing their properties for biotechnological applications, such as developing new antiviral strategies or using satellite DNAs as tools for gene silencing or other genetic manipulations. Sequence analysis could also track the spread and evolution of plant viruses and their associated satellite DNAs, providing insights into the dynamics of viral populations and the emergence of new strains. This manuscript was based on the insight of the significance of plant viral genome characterization research, and provided a full demonstration of the identification tools that was utilized.
Author Response
Comment: In this manuscript, the authors proposed the full-length viral component and three satellite molecules (PepLCYnV, ToLCCNB) linked to lettuce-infecting Begomovirus first reported in Yunnan, China. Methodology essentially included enrichment of circular DNA by RCA, followed by restriction enzyme digestion, PCR, cloning, and DNA sequencing. Meanwhile, PepLCYnA, an undiscovered alphasatellite was also proposed. It was believed to be the first discovery of lettuce as a new natural host of Begomovirus in China.
Understanding the genetic makeup of plant viruses and satellite DNAs could provide imminent opportunities for harnessing their properties for biotechnological applications, such as developing new antiviral strategies or using satellite DNAs as tools for gene silencing or other genetic manipulations. Sequence analysis could also track the spread and evolution of plant viruses and their associated satellite DNAs, providing insights into the dynamics of viral populations and the emergence of new strains. This manuscript was based on the insight of the significance of plant viral genome characterization research, and provided a full demonstration of the identification tools that was utilized.
Our response:Thank you very much for your comments.

Reviewer 2 Report
Comments and Suggestions for Authors
Comments and Suggestions for Authors:
The work submitted by Tang et al., "Identification and Genome Characterization of a Begomovirus and Satellite Molecules Associated with Lettuce (Lactuca sativa L.) leaf curl disease," reports a new isolate of PepLVYnV, a new betasathelite isolate, ToLCCNB, and two alphasatelite YN-2023-WJ-alpha1 and YN-2023-WJ-alpha2. YN-2023-WJ-alpha2 is proposed to belong to a new geminialphasatellite species, and the authors name it pepper leaf curl Yunnan alphasatellite (PepLCYnA).
This is the first report of PepLVYnV, ToLCCNB, AYVA, and PepLCYnA in L. sativa, a significant finding that expands our understanding of Begomovirus. The identification of L. sativa as a new host for Begomovirus is a noteworthy discovery, as it could have implications for the management of leaf curl disease in this plant species.
In general, the authors did very good and methodic work, but they should consider making a few changes to improve the quality of the paper.
Abstract:
Consider removing the accession numbers from the abstract to enhance readability and make it more accessible to a broader audience.
Minor changes in the wording to consider:
L1 replace exhibiting for showing
L1,2 replace discovered by found
L4 remove subsequently
L7, the authors could use 'exhibited' instead of 'shared'. Similarly, in L9 and L10,'displayed' and 'showed' could be used instead of 'shared'. This will help avoid repetition and improve the overall flow of the abstract.
Introduction:
The introduction, while well-structured and written, could be further enriched by including a more detailed review of geminivirus molecular genomic diversity at the species level and below. This would provide the audience with a more comprehensive understanding of the study's context.
Minor changes to consider:
L3 remove "etc" and another part of the text if apply
L11 It "It is thus clear that the" to Thus, the
L15 remove "hence"
L16 “depengding” to depending
Results
The findings are well-structured and organized, featuring well-described figures that enhance understanding.
There is no description in section 2.3 related to the differential amplification of the AYVA fragment (it should also be discussed in the Discussion section).
Minor changes to consider:
L1 "Some" to some
L5 "and an" to and
Materials and methods
4.1 Field investigation and Sample collection
It would be helpful to describe the coordinates of the location where the samples were collected.
4.3 Cloning of begomoviral genome and satellite molecules.
The fact that the RCA reaction was digested only with BamH I and EcoR I, while no other restriction enzymes were used, raises the possibility that other geminiviruses could be present but not cloned. This possibility was not discussed.
Discussion
In the discussion, it would be beneficial to explore the possibility of other geminiviruses and hypothesize how this viral complex might be encapsulated and transmitted. This would engage the audience and make the discussion more comprehensive.
Conclusion
The conclusions effectively summarize the study's findings and their alignment with the stated objectives. This alignment underscores the thoroughness and accuracy of the research.
Author Response
In general, the authors did very good and methodic work, but they should consider making a few changes to improve the quality of the paper.
Comment1:Abstract: Consider removing the accession numbers from the abstract to enhance readability and make it more accessible to a broader audience.
Minor changes in the wording to consider:
L1 replace exhibiting for showing
L1,2 replace discovered by found
L4 remove subsequently
L7, the authors could use 'exhibited' instead of 'shared'. Similarly, in L9 and L10,'displayed' and 'showed' could be used instead of 'shared'. This will help avoid repetition and improve the overall flow of the abstract.
Our response: We had removed the accession numbers, and revised in the wording according to the suggestion.
Comment2:Introduction: The introduction, while well-structured and written, could be further enriched by including a more detailed review of geminivirus molecular genomic diversity at the species level and below. This would provide the audience with a more comprehensive understanding of the study's context.
Minor changes to consider:
L3 remove "etc" and another part of the text if apply
L11 It "It is thus clear that the" to Thus, the
L15 remove "hence"
L16 “depengding” to depending
Our response: We had revised in the wording according to the suggestion.
Comment3:Results: The findings are well-structured and organized, featuring well-described figures that enhance understanding. There is no description in section 2.3 related to the differential amplification of the AYVA fragment (it should also be discussed in the Discussion section).
Minor changes to consider:
L1 "Some" to some
L5 "and an" to and
Our response: We had revised in the wording according to the suggestion.
Regarding the differential amplification of the AYVA fragment, the result in MS indicated that AYVA could be detected in these samples, and the differential amplification did not affect the conclusion, hence it was not discussed in the Discussion section.
Materials and methods:
Comment4:4.1 Field investigation and Sample collection
It would be helpful to describe the coordinates of the location where the samples were collected.
Our response: We had added the coordinates of the location where the samples were collected.
Comment5:4.3 Cloning of begomoviral genome and satellite molecules.
The fact that the RCA reaction was digested only with BamH I and EcoR I, while no other restriction enzymes were used, raises the possibility that other geminiviruses could be present but not cloned. This possibility was not discussed.
Our response: We had added relevant content in the Discussion section.
Comment6:Discussion: In the discussion, it would be beneficial to explore the possibility of other geminiviruses and hypothesize how this viral complex might be encapsulated and transmitted. This would engage the audience and make the discussion more comprehensive.
Our response: We had added relevant content in the Discussion section.
Comment7:Conclusion: The conclusions effectively summarize the study's findings and their alignment with the stated objectives. This alignment underscores the thoroughness and accuracy of the research.
Our response: Thank you very much for your comment.
Thank you very much for your comments and corrections. We have revised the MS according to your suggestions. Your suggestions are really valuable in improving the quality of our MS.
